

# Stochastic processes dominate the community assembly of ectomycorrhizal fungi associated with *Betula platyphylla* in Inner Mongolia, China

Min Li[1,2,*], Zhaoyun Meng[1,*], Jinyan Li[3], Xuan Zhang[4], Yonglong Wang[4], Xinyu Li[5], Yuze Yang[1], Yue Li[1], Xunjue Yang[1], Xiuli Chen[4] and Yongjun Fan[3]

[1] College of Life Science and Technology, Inner Mongolia Normal University, Huhhot, China
[2] Key Laboratory of Biodiversity Conservation and Sustainable Utilization for College and University of Inner Mongolia Autonomous Region, Hohhot, China
[3] School of Life Science and Technology, Inner Mongolia University of Science and Technology, Baotou, China
[4] Faculty of Biological Science and technology, Baotou Teacher's College, Baotou, China
[5] School of Energy and Environment, Inner Mongolia University of Science and Technology, Baotou, China
[*] These authors contributed equally to this work.

Corresponding author
Yongjun Fan, fanyj1975@163.com

## ABSTRACT

The maintenance and driving mechanisms of microbial community structure have become important research focuses in microbial ecology. Therefore, clarifying the assembly of ectomycorrhizal (EM) fungal communities can provide a relevant basis for studying forest diversity, ecological diversity, and ecological evolution. *Betula platyphylla* is a typical EM dependent tree species with characteristics such as renewal ability and strong competitive adaptability, and it plays a crucial ecological function in Inner Mongolia. However, the research on EM fungi's diversity and community assembly is very limited. We investigated EM fungal communities associated with *B. platyphylla* from 15 rhizosphere soil samples across five sites in Inner Mongolia. The fungal rDNA ITS2 region was sequenced using Illumina Miseq sequencing. A total of 295 EM fungal OTUs belonging to two phyla, three classes, nine orders, 20 families, and 31 genera were identified, of which *Russula*, *Cortinarius*, and *Sebacina* were the most dominant taxa. Significant differences existed in the composition of dominant genera of EM fungi across the five sites, and the relative abundances of dominant genera also showed significant differences among the sites. The β NTI and NCM fitting analyses suggest that stochastic processes mainly determine the EM fungal community assembly. Our study indicates that *B. platyphylla* harbors a high EM fungal diversity and highlights the important role of the stochastic process in driving community assembly of mutualistic fungi associated with *B. platyphylla* in north China.

## INTRODUCTION

Soil microorganisms are an important component of soil ecosystems, driving and influencing many ecosystem processes such as organic matter decomposition, nutrient

cycling, and ecosystem productivity. They are crucial for maintaining global ecosystem functions (*Tedersoo et al., 2014*; *Crowther et al., 2019*). Mycorrhizal fungi account for approximately 70% of the total microbial population in soil ecosystems, making them one of the most important functional groups in the ecosystem (*Crowther et al., 2019*). EM fungi are one of the main types of mycorrhizal fungi, which could form symbiosis with 30 lineages of terrestrial plants. The EM plants (*e.g.*, Pinaceae, Fagaceae, Betulaceae, and Salicaceae) played important roles in the forest ecosystems (*Tedersoo, Bahram & Zobel, 2020*), which, clarifying the underlying factors of EM fungi community assembly of these trees species could vastly improve our predictability in the areas of sustainable forests, biochemical dynamics, and ecological processes under several global changes (*Wang et al., 2021*). An in-depth understanding of soil fungal diversity, composition characteristics of different functional groups, and their impact mechanisms is important for soil health management, sustainable development of ecosystems, and predicting the response and feedback mechanism of microbial communities under changes in environmental factors (*Tedersoo et al., 2014*; *Bardgett & Putten, 2014*).

Several studies have attempted to investigate the EM fungal community's drivers from environmental filtering (*i.e.,* plant, soil, and climate) and dispersal limitation (*i.e.,* spatial distance). For example, research on the geographical distribution of soil fungi on a global scale has shown that climate factors, soil characteristics, and spatial patterns are the best predictive factors for soil fungal richness and community composition (*Zhou & Ning, 2017*). Tree species characteristics are important determinants for EM fungal community assembly (*Otsing et al., 2021*). A large number of studies have shown that soil pH is an important predictor of microbial diversity in response to global change factors (*Wang et al., 2015*). Studies indicate that temperature is the most important regulatory factor for microbial diversity in forest soil at a large spatial scale (*Zhou et al., 2016*). It has gradually become a trend to comprehensively understand the mechanism of community construction from different perspectives by combining neutral theory and niche theory. Some studies have shown that the construction of microbial communities is mainly influenced by deterministic (niche theory) and stochastic processes (neutral theory) (*Chen et al., 2020*; *Zhang et al., 2021*) *e.g.*, *Liu et al. (2021)* found that the assembly of rhizosphere fungal community associated with *Pinus massoniana* in East Sichuan was determined by a deterministic process. *Wang et al. (2021)* emphasized a determinant role of dispersal limitation in the stochastic process on EM fungal community assembly associated with common pine species in semiarid and cold temperate forests in Inner Mongolia of China. However, different fungal groups exhibit significant differences in their patterns with changes in climate, soil, and plant parameters (*Tedersoo et al., 2014*). In general, the effect of the deterministic process in defining the EM fungal community could be attributed to the selection of biotic and abiotic environmental variables on EM fungi through their fitness in response to the surrounding conditions (*De Wit & Bouvier, 2006*). These studies have clarified the different ecological processes and their relative importance in controlling some microbial communities. However, there is still an important question regarding the ecological processes underlying the community assembly of EM fungi, and the question

remains uncertain and warrants further exploration, considering crucial role of the process of community assembly in ecosystems.

*Betula platyphylla*, as a pioneer species in the succession process of natural secondary forest ecosystems plays a crucial role in temperate deciduous broad-leaved forests. It is of great significance in maintaining regional ecological balance. *B. platyphylla* is a typical EM-dependent tree species. It has been reported that EM fungi associated with *B. platyphylla* mainly include Russulaceae, Amanitaceae, Boletaceae, Cortinariaceae, Tricholomataceae, and other families, as well as *Gomphidius, Suillus, Pisolithus, Boletus, Cortinarius, Inocybe, Sebacina, Piloderma, Hebeloma, Cenococcum, Tuber, Geopyxis,* and *Amanita, etc* (*Bai et al., 2006*; *Fan & Yan, 2013*). The findings mentioned above were primarily based on the identification of EM morphology. However, relying solely on EM morphology to identity EM fungi poses several challenges. Firstly, it brings a heavy workload and low efficiency. Secondly, it cannot accurately reflect EM fungal diversity because some EM fungi may not form EM root tips when sampling collections. Therefore, to better understand the structure of the EM fungal community of *B. platyphylla*, *Yang, Yan & Wei (2018)* have studied the composition of the EM fungal community of *B. platyphylla* using high-throughput sequencing technology in Heilihe Natural Reserve, Saihanwula Natural Reserve and Helanshan Natural Reserve of Inner Mongolia (*i.e.,* a local scale). The results showed that *B. platyphylla* has a high diversity of EM fungi. In summary, the past studies on the mycorrhizal fungi of *B. platyphylla* mainly focuses on diversity of EM fungal, and the classification and identification of fungi. A number of studies have shown that there are significant differences in EM fungal communities among various sites, even for the single plant species, which mirrors the site/geographic effect on EM fungal communities (*Hackel et al., 2022*). We still know less about the EM fungal community structure, and the process of the community assembly of ectomycorrhizal fungi associated with *Betula platyphylla* at a large scale in the Inner Mongolia. Therefore, this study investigated and researched EM fungi of *B. platyphylla* at a large scale (spanning about 2,400 km from east to west and 1,700 km from north to south). There is a significant zonal distribution of water, heat, and vegetation across the sites from east to west in the Inner Mongolia. In terms of climate and temperature zones, it successively crosses the cold, medium, and warm temperate zones, and warm temperate zone. From the perspective of humidity, it presents a climate change characteristic of moist, semi-humid, and semi-arid in sequence. So, this paper intends to answer the following questions: (1) Does EM fungal diversity and community composition change across environmental zones? (2) What is the relative importance of environmental filtering *versus* dispersal limitation in influencing EM fungal community assembly?

## MATERIALS & METHODS

### Site description and sampling

Five typical secondary forests of *B. platyphylla* were selected over a 2,300-km west–east transect across Inner Mongolia in north China (Fig. 1), The selected forests spanned from the eastern temperate continental monsoon climate zone to the western temperate continental climate zone. According to the climate data retrieved from the WorldClim

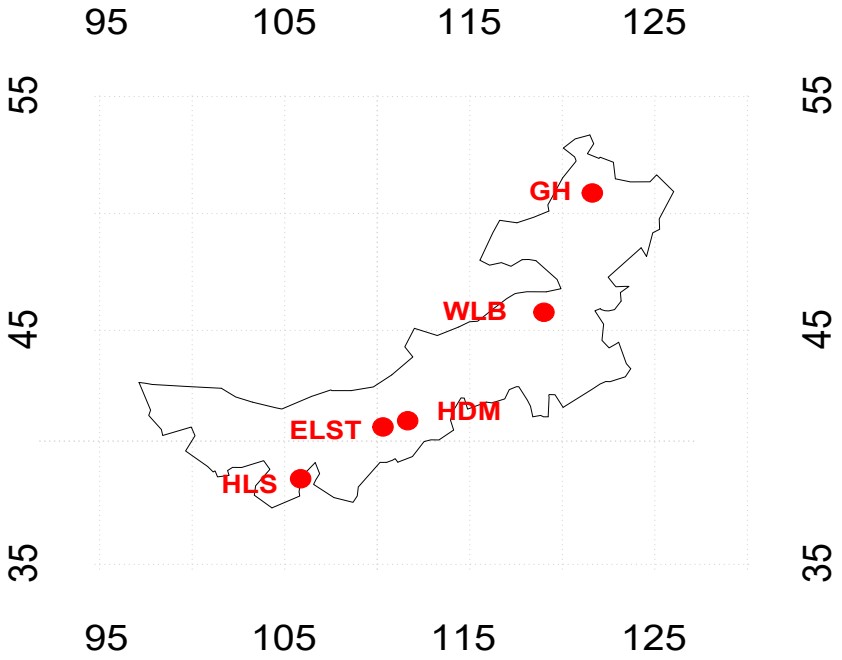

**Figure 1  Sampling site map.** Five typical sampling sites for rhizosphere soil of *Betula platyphylla* in Inner Mongolia Autonomous Region, China. GH, Genhe; WLB, Wulanba National Nature Reserve; HDM, Hademen National Forest Park; ELST, Erlongshitai National Forest Park; HLS, Helan Mountain National Nature Reserve.

dataset with a 30-arc-second resolution, the mean annual temperature (MAT) within this area varied from 4.75 to 5.97 °C, and the mean annual precipitation (MAP) ranged from 267 to 506 mm. (*Hijmans et al., 2005*). The forests were approximately 20 years old and had no disturbance. Five sample sites from west to east are Helan Mountain National Nature Reserve (HLS), Erlongshitai National Forest Park (ELST), Hademen National Forest Park (HDM), Wulanba National Nature Reserve in Inner Mongolia (WLB), and Genhe (GH) (Table S1) and field experiments were approved by the Helan Mountain National Nature Reserve in Inner Mongolia. At each sampling plots, nine tree individuals were selected from each pure birch forest, select three tree individuals at each altitude and the individuals were >10 m away from each other to ensure sample independence (*Lilleskov et al., 2004*). Three rhizosphere soil cores were collected from three individuals and mixed as one composite sample (~200 g). The fresh soil samples were sieved through a two mm sieve to remove the roots and debris. Subsamples used for soil physical and chemical properties tests were stored at 4 °C prior to the analyses, and the subsamples for DNA extraction were frozen at −80 °C.

## Analysis of soil properties

Soil physical and chemical properties including soil total nitrogen (TN), total organic carbon (TOC), available potassium (AK), available phosphorus (AP), soil water content (SWC), pH value, and soil electrical conductivity (EC) were analyzed in this study.

According to agricultural standards, the test were conducted directly using a soil nutrient analyzer (LASA AGRO 1900, STEPS, Germany).

### Rhizosphere soil DNA extraction, PCR, and MiSeq sequencing

Genomic DNA was extracted from 0.25 g frozen soil using the PowerSoil DNA isolation kit (Mobio Laboratories, Inc.), according to the manufacturer's instructions. The detailed method for DNA extraction can refer to the method described by *Gao et al. (2013)*, and other specific methods such as the PCR protocol available by *Ihrmark et al. (2012)*, *Wang et al. (2019a)* and *Wang et al. (2019b)*. Briefly, the fungal internal transcribed spacer 1(ITS2) region was amplified using the PCR primers of ITS1F and ITS2. The PCR products of each sample were purified and mixed at equimolar amounts (200 ng) and then sequenced on an Illumina MisSeq PE250 platform (Illumina, San Diego, CA, USA). High throughput sequencing work was completed by Shanghai Personal Biotechnology Co., Ltd (Shanghai, China).

### Bioinformatics analysis

The raw data used QIIME Pipeline-Version 1.7.0 (*Caporaso et al., 2010*) for quality control. Eliminate the sequence of >6 ambiguous bases and invalid primers; secondly, ITSx software (*Bengtsson-Palme et al., 2013*) was used to extract the ITS2 region of the fungi. The Usearch v.11 was used to check chimera, and command to remove the potential chimera sequence (*Peay, Garbelotto & Bruns, 2010*). Based on the high-quality non-chimera ITS2 sequence obtained from the above process, the Usearch v.11 was used to perform the operational taxonomic units (OTU) assignment according to the sequence similarity of 97%. The representative sequence of each OTU (the largest number of sequences) is compared with BLAST (basic local alignment search tool) using INSDC (international nucleotide sequence databases collaboration) and UNITE as reference databases (*Altschul et al., 1990*). Use the identification standard of *Tedersoo et al. (2014)* as a reference for the identification of fungal OTU, that is, according to the similarity value greater than 90%, 85%, 80%, and 75%, the fungal OTU can be identified to the classification level of genus, family, order, and class. According to the nomenclature method of *Tedersoo & Smith (2013)*, EM fungi are identified if the OTUs were highly similar to known EM fungi sequences and the lineages were also identified. In order to avoid the influence of the difference in the number of sequences between samples on the subsequent analysis results, the EM fungal data using the subsample command the rarefy function in the ape package (*Paradis, Claude & Strimmer, 2004*) in R v3.6.1.The raw sequence data reported in this paper have been deposited in the Genome Sequence Archive in National Genomics Data Center, China National Center for Bioinformation/Beijing Institute of Genomics, Chinese Academy of Sciences (GSA: CRA013683) that are publicly accessible at CRA013683.

### Statistical analysis

All statistical analysis was performed in R 4.3.1 (*R Core Team, 2022*). EM fungal OTU accumulation curves for each site were drawn using the specaccum command in the vegan package (*Oksanen et al., 2013*). Alpha diversity including OTU richness, Shannon-Wiener, Simpson, Chao1, and abundance-based coverage estimator (ACE) index were calculated

for EM fungal analysis in the vegan package. Beta diversity for EM fungal analysis including the principal coordinate analysis (PCoA), which was based on Bray–Curtis dissimilarity matrices, was used to visualize the differences in EM fungal communities at five sites, and the ordiellipse function was used to fit the 95% CIs of sites onto the PCoA ordination. Subsequently, permutational multivariate ANOVA (PERMANOVA) with 999 permutations was adopted to evaluate the significance of the difference in the EM fungal communities among the five sites. The nonmetric multidimensional scaling (NMDS) analysis was finished with the metaMDS command in the vegan package.

To ascertain the potential significance of stochastic processes in the assembly of EM fungal communities, the Niche Conservatism Model (NCM) was utilized to account for the relationship between OTU detection frequency and relative abundance. In recent years, phylogeny and null model analysis have been widely employed to comprehend the relative importance of deterministic and stochastic processes in community assembly (*Stegen et al., 2012*). In accordance with the method detailed in the study by *Stegen et al. (2012)*, the β-mean nearest taxon distance (βMNTD) was applied to calculate the phylogenetic turnover between samples. Moreover, based on the standardized estimate β-nearest taxon index (βNTI) of MNTD in the picante packages, the standard deviation of the observed βMNTD value from the null distribution of βMNTD was computed. βNTI values greater than 2 or less than −2 signify a deterministic process, which mainly encompasses heterogeneous selection and homogeneous selection. |βNTI| values less than 2 represent stochastic processes, including homogenizing dispersal, dispersal limitation, and drift (*Liu et al., 2020*).

## RESULTS

### Ectomycorrhizal fungal database summary

A total of 377,969 sequences were obtained after de-redundancy and de-chimera processing. These sequences were classified into 2,039 OTUs based on the 97% similarity level, and after identification, 1,132 OTUs (331,960) were identified as fungi. Finally, 295 EM fungal OTUs were obtained through the identification and rarefaction of EM fungi, which is carried out for the analysis of fungal community composition (Table S2). Among the 295 OTUs, 22 OTUs belonged to Ascomycota (3.01% of total EM fungal reads), 273 OTUs to Basidiomycota (96.99%).

### Ectomycorrhizal fungal diversity

The accumulation curves of each site did not show any signs of reaching an asymptote, suggesting that further sample collection may result in unknown EM fungal OTUs (Fig. 2A). We analyzed the alpha diversity of the EM fungal community of *B. platyphylla* in five sites, and an ANOVA analysis and nonparametric Wilcoxon test were used to calculate the significant difference of species richness between different sites ($P < 0.05$). The nonparametric Wilcoxon test showed that EM fungal OTUs' richness (log transformed), Shannon and Chao1 indices significantly differed across five sites, and were $55.600 \pm 6.807$ (mean $\pm$ SE), $3.808 \pm 0.261$ (mean $\pm$ SE) and $68.575 \pm 31.397$ (mean $\pm$ SE), respectively. The HSD turkey tests further indicated that the indices of ELST were significantly lower

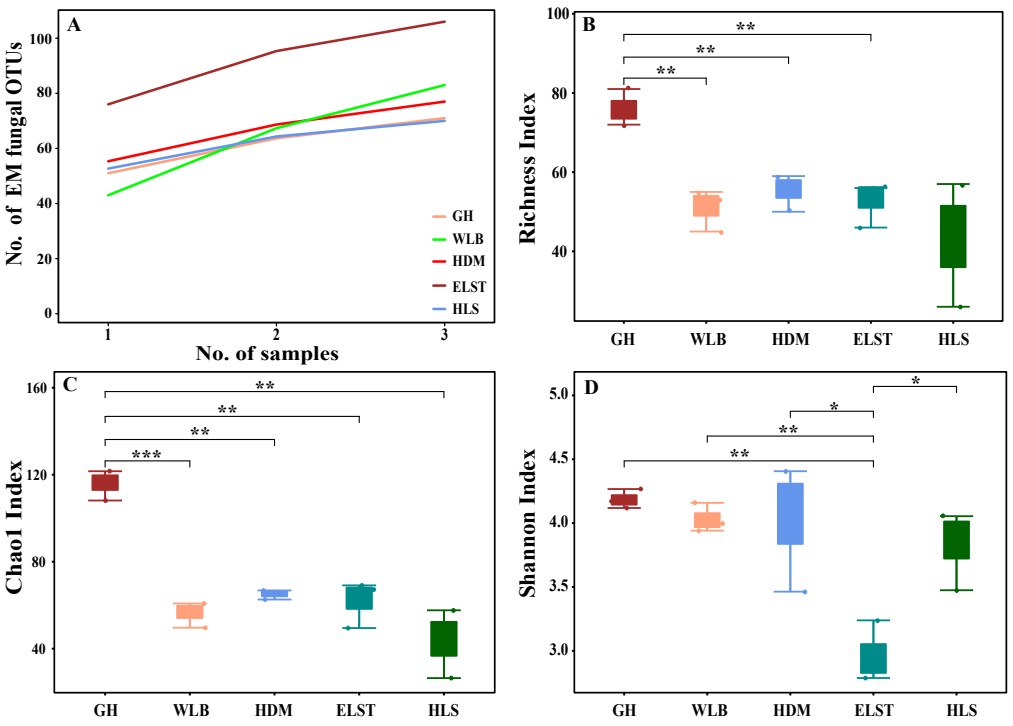

**Figure 2** **Accumulation curves of each site and α-diversity.** Ectomycorrhizal (EM) fungal diversity. Accumulation curves of EM fungal operational taxonomic units (OTUs) (A), EM fungal OTUs richness (B), Chao1 index (C) and Shannon index (D) at five sites. According to adonis test, significance: *, $P < 0.05$, **, $P < 0.01$, ***, $P < 0.001$. GH, Genhe; WLB, Wulanba; HDM, Hademen; ELST, Erlongshitai; HLS, Helanshan.

than those in GH, HDM, HLS and WLB (Wilcoxon: $p < 0.05$, Figs. 2B–2D). The venn diagram showed that the number of OTUs was different in the five sites, from west to east HLS-83, ELST-70, HDM-77, WLB-71, and GH-106, respectively. The proportion of OTUs number in different regions is 28.14%, 23.73%, 26.10%, 24.07%, and 35.93% of total OTUs number, respectively, indicating significant differences among fungal communities in different regions, with HLS and GH having the higher number of OTUs, and of which only one OTU were shared by the five sites. Furthermore, each of the sites harbors unique OTUs, 36, 19, 3, 39 and 69 fungal OTUs only existed on HLS, ELST, HDM, WLB, and GH, respectively, accounting for 56.27% of total OTUs number (Fig. 3). The above results indicated that there were significant differences in the diversity of EM fungi in the five sites, and multiple comparisons showed that the diversity of EM fungi in GH was significantly higher than that in the other four sites (Table S1).

## Ectomycorrhizal fungal community composition

A total of 30 lineages were found in 15 soil samples of five sites in the current study, of which *Russula-Lactarius* (26.65%), *Sebacina* (18.66%), *Cortinarius* (17.44%), *Inocybe* (9.25%), *Hygrophorus* (8.3%), *Amanita* (4.49%), *Tomentella-Thelephora* (3.83%), *Clavulina* (3.02%), *Piloderma* (2.71%) and *Terfezia-Peziza* (1.51%) were the dominant evolutionary

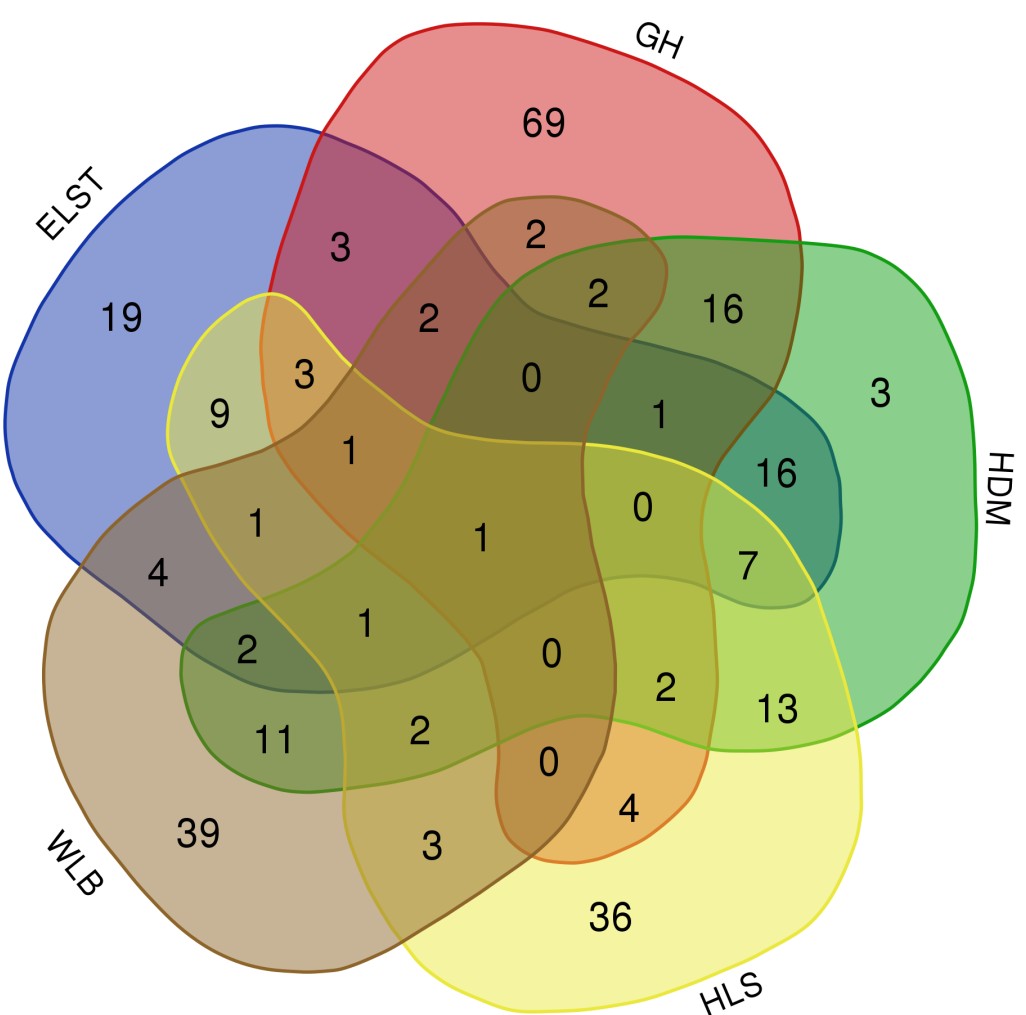

**Figure 3** **The Venn diagram of EM fungi OTUs number in the rhizosphere of *B. platyphylla* from five sites in Inner Mongolia.** GH, Genhe; WLB, Wulanba; HDM, Hademen; ELST, Erlongshitai; HLS, Helan­shan.

lineages (Table S3, >1% of total sequences). The EM fungal community composition was different in different sites, *Wilcoxina*, *Laccaria*, *Tomentellopsis* and *Serendipita1* were only detected at GH; *Cantharellus* were only detected at WLB; *Ceratobasidium3* and *Tuber-Helvella* were only detected at HDM; *Otidea* and *Genea-Humaria* were only detected at ELST; *Paxillus-Gyrodon* and *Marcelleina-Peziza* were only detected at HLS. Comparing the relative abundance of the top 10 EM fungal lineages in the five sites, we found that *Russula-Lactarius*, *Sebacina* and *Cortinarius* (>10% of total sequences) were the most abundant lineages of EM fungi (Fig. 4).

The EM fungal community structure with the different sites was analyzed using PCoA based on the Bray–Curtis distance (Fig. 5). PCoA ordination analysis showed that the PCoA first-axis explanation rate was 24.46%, the second-axis explanation rate was 22.36%, in which clear area separation in terms of sampling sites. Moreover, PERMANOVA agreed

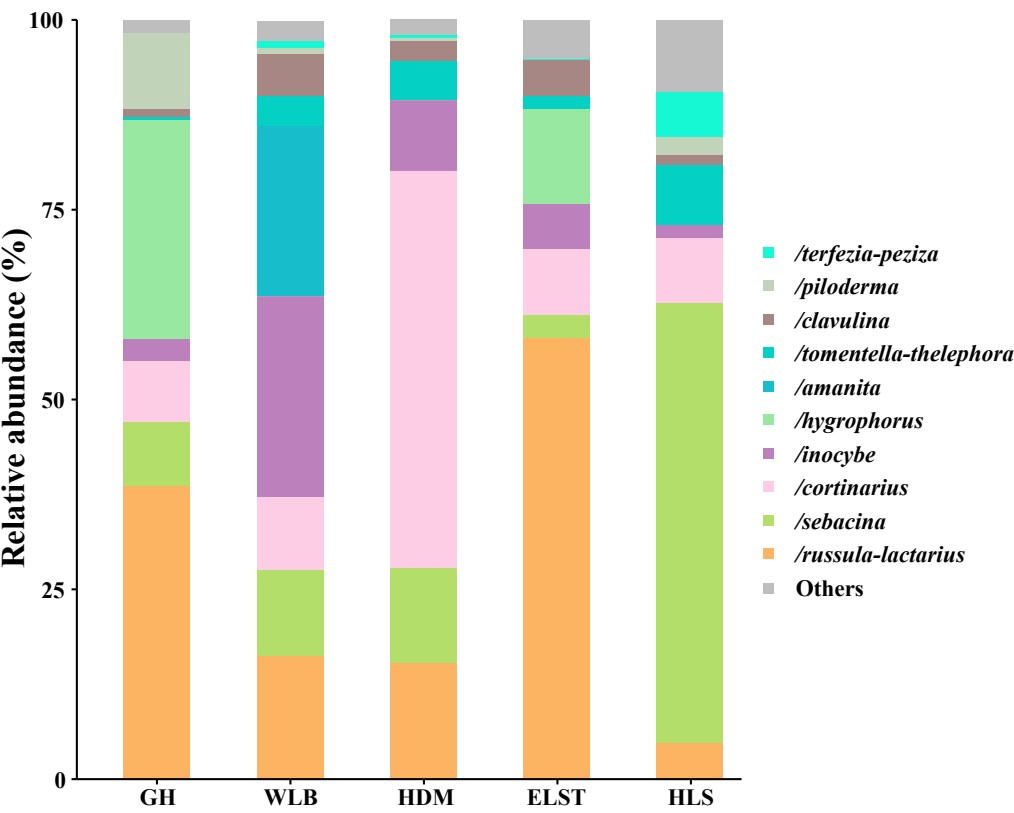

**Figure 4 Ectomycorrhizal fungal lineage and their relative abundance in five sites.** Here, we only showed abundant lineages (>1% of total sequences). GH, Genhe; WLB, Wulanba; HDM, Hademen; ELST, Erlongshitai; HLS, Helanshan.

with the PCoA in that there were significant differences in the EM fungal communities between different sampling sites (Fig. 5, Adonis: $R^2 = 0.89$, $P = 0.001$). NMDS ordination revealed that EM fungal communities of the five sites were clearly separated, particularly the community in GH (Fig. 6). In addition, an environmental fitting test demonstrated climate conditions, and soil properties were significantly correlated with the EM fungal community (Fig. 6 and Table S4).

## Ectomycorrhizal fungal community assembly processes

The EM fungal community ecological process was fitted to the NTI and βNTI, the both NTI and βNTI values were used to assess the different ecological assembly processes in EM fungal communities. The results show that the NTI values ranged from −1.8 to 0.22, and the mean NTI value (0.52) across all samples was not significantly different from 0 ($P > 0.05$, Fig. 7A), indicating the phylogenetic relatedness in the communities was stochastic. Most βNTI values fall in the range of −2 to 2 (87.74%; Fig. 7B), while the case of |βNTI|>2 is only 12.26%, which also indicates that the assembly process of EM fungal communities is mainly affected by stochastic processes. Meanwhile, the result of calculating the distribution of |RC$_{bary}$| shows ecological drift, dispersal limitation and homogeneous dispersal account

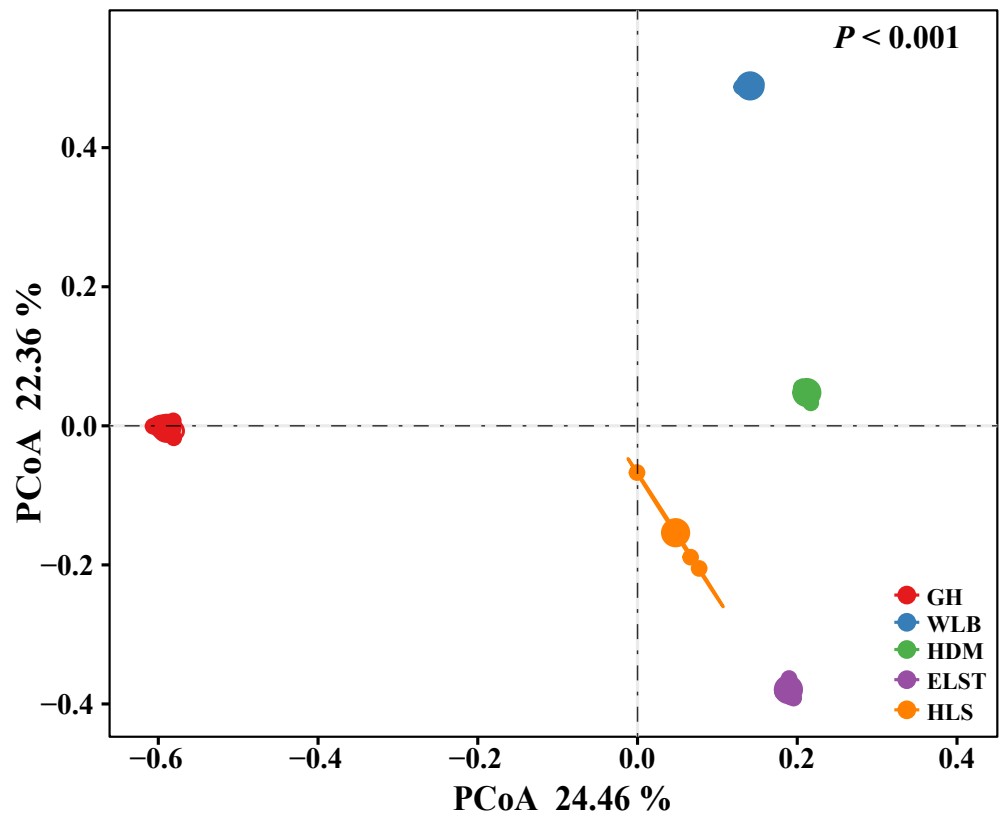

**Figure 5** Principal Coordinates Analysis (PCoA) ordering of ectomycorrhizal fungal community composition based on Bray–Curtis distance(*P* < 0.001). GH, Genhe; WLB, Wulanba; HDM, Hademen; ELST, Erlongshitai; HLS, Helanshan.

for 42%, 42% and 16% respectively (Fig. S1), indicating that stochastic processes had a stronger effect on the community assembly of EM fungi than the deterministic processes in the this study.

## DISCUSSION

Our study indicating that there are abundant EM fungal communities in the rhizosphere of *B. platyphylla*. Previous studies have shown that forests with a high fungal diversity commonly harbor a strong ability to resist ecosystem disturbances and remediation (*Guo et al., 2018*), and have shown a significant positive correlation between soil biodiversity and various ecosystem functions (nutrient cycling, decomposition, plant production, and reducing potential pathogenicity) (*Delgado-Baquerizo et al., 2020*). The abundant EM fungal community in the rhizosphere of *B. platyphylla* plays an important role in maintaining the stability of the forest ecosystem and nutrient cycling, which may be an important reason for its widespread distribution in northern China.

The five sites in this study are approximately 2,000 km from east to west, with significant differences in climate, geography, soil, and other aspects. The genera of *Russula*, *Cortinarius*,

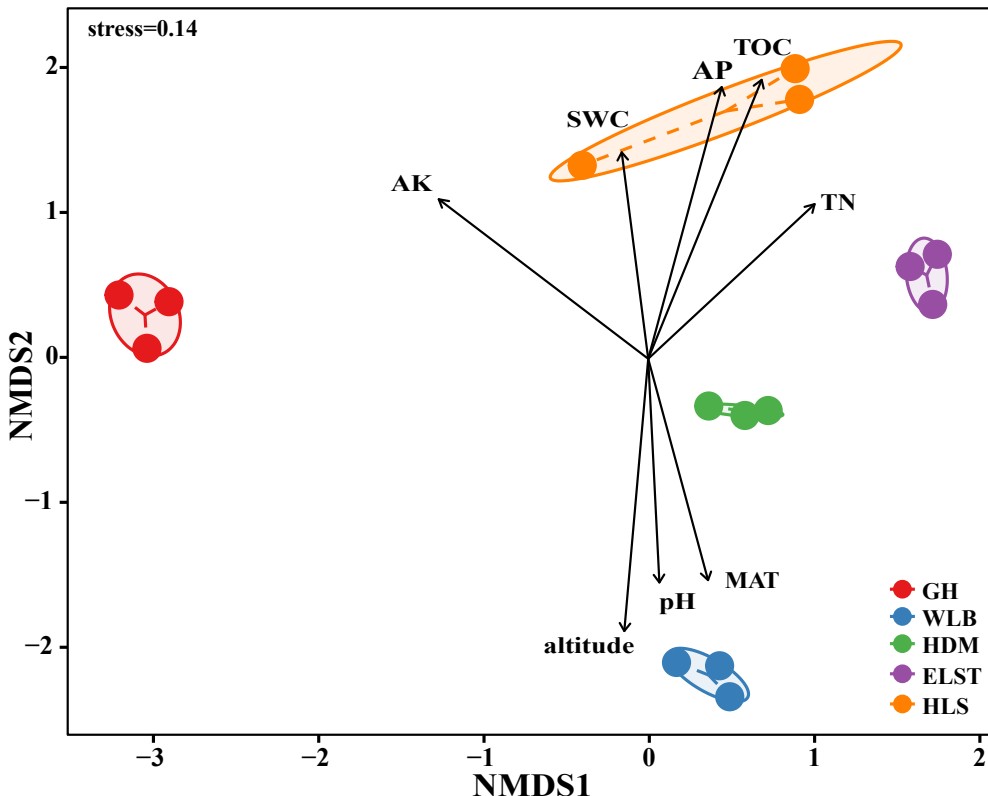

**Figure 6** **Non-metric multidimensional scaling (NMDS) of the EM fungal community composition (Bray–Curtis distance) (stress = 0.14). Ellipses indicate 95% confidence intervals around centroids for each sites.** Significant spatial, soil and climatic variables were fitted onto the NMDS ordination. TN, soil total nitrogen; TOC total organic carbon; AK, available potassium; AP, available phosphorus; SWC, soil water content; pH; altitude; MAT, mean annual temperature. GH, Genhe; WLB, Wulanba; HDM, Hademen; ELST, Erlongshitai; HLS, Helanshan.

*Sebacina*, *Inocybe*, and *Hygrophorus* are all distributed in the five sites. This is consistent with the conclusions of other researchers. Their results showed that *Russula* and *Hymenomyces* are widely distributed in temperate regions and can coexist with a variety of trees or shrubs (*Jang & Kim, 2012*; *LeDuc et al., 2013*). They are the dominant EM fungi of *B. platyphylla*. It has been reported that the relative abundances of these two genera are linearly correlated (*Xing et al., 2020*). Previous studies have shown that *Cortinarius* can secrete a large amount of peroxidase (*Bödeker et al., 2014*), which may play an important role in degrading dead branches and leaves. In addition, *Sebacina* is also widely distributed in various forest ecosystems, with almost no host specificity (*Oberwinkler et al., 2012*).

At the same time, our study found that the community structure of EM fungi in the five sites differed to some extent, and the dominant genera and their relative abundance in each site differed. *Russula* has the highest relative abundance in the eastern GH and central ELST of Inner Mongolia. *Cortinarius* is the dominant genus in the HDM of central Inner Mongolia, while *Sebacina* is the dominant genus in the HLS of western Inner Mongolia.

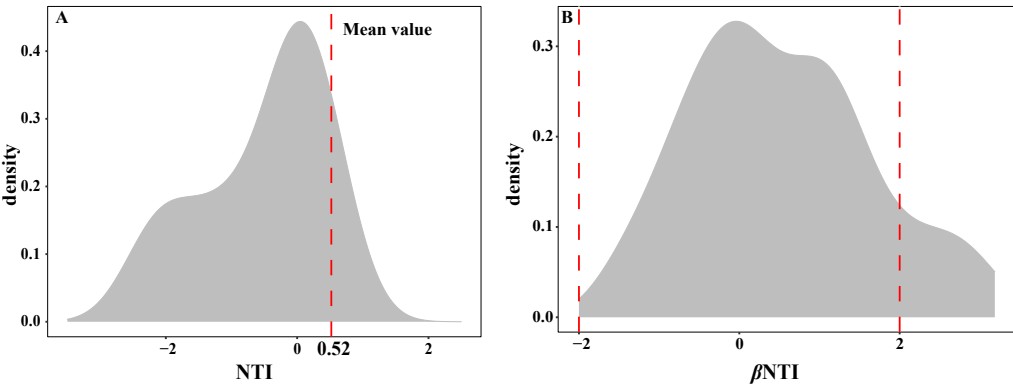

**Figure 7   Ectomycorrhizal fungal community ecological process.** Ecological processes underlying community assembly of ectomycorrhizal (EM) fungi. (A) Distribution of nearest taxon index (NTI); (B) beta nearest taxon index ($\beta$NTI).

*Otidea* only exists in ELST. Each site has its unique species, and there are also species shared by two or more sites, which better illustrates the diversity and differences between fungal communities. This is because the five sites in this study have a great span in geographical distance, and there are significant differences in ecological environmental factors among different sites, while a large number of studies show that the community structure of EM fungal is affected by multiple environmental factors (*Tedersoo et al., 2014*; *Crowther et al., 2019*). In addition, it is also related to the different microenvironments of EM fungi in different sites. Although the constructive species in each sample in this study are pure birch forest, the complexity of litter resources available to EM fungi is different due to different associated shrubs and herbs under the forest (*Urbanová, Šnajdr & Baldrian, 2015*), leading to certain differences in their community structure. Other studies also believe a group of relatively abundant fungi exists in a specific environment (*Fitzpatrick et al., 2020*). In terms of geographical location, GH is located in a low altitude area, and is also the easternmost area among the five sites, which is closer to the coast compared to other sites. Regarding environmental factors, GH is affected by the southeast monsoon, which brings a large amount of water vapor into the sea surface, resulting in the highest MAP, lowest MAT, and the largest proportion of SWC. Therefore, there is a significant difference in microbial communities compared to inland areas. Due to the close distance and the small differences in soil rhizosphere environment between ELST and HDM, the natural conditions are the same, resulting in smaller community differences. After comparing ELST with HLS and HDM, the difference between HLS and HDM is greater due to the geographical location of HLS is closer to inland areas, resulting in lower temperatures and less precipitation. The significant differences in soil microbial communities between ELST and GH are mainly due to different ecological environments.

In this study, all factors including TN, TOC, AK, pH, longitude, and latitude significantly impact on the distribution of EM fungi. The soil microbial community is a dynamically changing self-organizing system, and its community structure and diversity are influenced

by various environmental factors, such as vegetation factors, climate factors, and soil factors.

Research has shown that pH value is an important factor affecting soil microbial diversity on a global scale (*Steidinger et al., 2019*). The analysis of the community structure of EM fungi in the rhizosphere soil of *B. platyphylla* at a large scale in this study also supports the above conclusions. It spans about 2,400 km from east to west in Inner Mongolia, and the soil pH value in the site shows a significant trend of change, gradually transitioning from the alkaline soil of the HLS in the western (pH 7.51–8.57) to the acidic soil of the GH in the eastern (pH 4.84–4.91). In addition, pH value is positively correlated with altitude, MAT, and EC, and negatively correlated with AK, SWC, longitude, latitude, AP, TOC, and TN. The research shows that these factors influence the community structure of EM fungi (*Miyamoto et al., 2015*). In this study, pH significantly impacts on the community composition of EM fungi. This may be because EM fungi produce a lot of organic acids in the process of mycorrhizal symbiosis with the host, which reduces the soil pH value, changes environmental factors (such as nutrient availability, organic carbon) and undergrowth vegetation structure, and then affects the community structure of EM fungi (*Hedwall et al., 2018*).

A large number of studies have shown that TN has a significant impact on the community structure of EM fungi, and the results of our study also support the above conclusions. TN may affect soil microbial community structure by affecting the diversity and dominance of aboveground vegetation (*Avis et al., 2003*). *Corrales et al. (2017)* suggest that to some extent, nitrogen deposition reduces the colonization of EM fungi and alters their community structure, and with the increase of nitrogen, the abundance of some EM fungal groups increases while some decreases. Soil water content is another important factor affecting soil microbial abundance on a global scale (*Steidinger et al., 2019*). The sampling site in this study has a great span in longitude, and the soil water conditions at each site are significantly different. Hence the community composition of EM fungi at each site is significantly different. Soil water content may affect the content of chemical ions in the soil. In contrast, the water was affected by various chemical process and biological processes in the soil, thus affecting the pH value of the soil. These findings suggest that different EM fungi taxa harbor distinct ecological niches and thus prefer to exist in certain habitats. Therefore, we may speculate that the distribution of EM fungal communities have been contributed to by the variation in the distribution and/or preference for the site of EM fungal OTUs. At same time, the variations in EM fungal communities and distributions/preference of EM fungal OTUs could be fundamentally attributed to the niche difference among fungi OTUs, because different OTUs preferred various habitats in terms of temperature, soil water, and nutrients *e.g.,* (*Glassman, Wang & Bruns, 2017*; *Miyamoto, Terashima & Nara, 2018*); for example, some EM fungal OTUs could adapt well to relative cold environments. In this study, GH has the most unique OTUs, and the richness and diversity of EM fungal OTUs are significantly higher than other sites, while the MAT of GH is significantly lower than other sites. The research results show that some EM fungi prefer to survive in lower temperature environments, possibly due to the high organic carbon content in cold habitats, which increases the diversity of EM fungi (*Rosling*

*et al., 2003*; *Buée et al., 2007*). Moreover, compared with other soil fungi, EM fungi have significant competitive advantages in nutrition and carbon acquisition in cold habitats (*Fernandez & Kennedy, 2016*). GH is located in a low altitude area with higher longitude and is also the easternmost area among the five sites, which is closer to the coast compared to other sites. The influence of the southeast monsoon on GH results in a large amount of water vapor being brought into the sea surface, resulting in the highest MAP, lowest MAT, and the largest proportion of SWC. Therefore, there are significant differences in microbial communities compared to inland areas.

Indeed, the difference in climatic conditions could result in distinct soil nutrients, which further drive EM fungi community variations, and common fungi species grow well under suitable soil nutrients and water content conditions, for example, the relative abundance of certain fungi decreases with an increase in pH value, while others exhibit opposite trends (*Wang et al., 2015*). Similarly, certain fungi groups increase with high nutrients levels, while others are inhibited (*Paungfoo-Lonhienne et al., 2015*). As is well known, different EM fungal species occupy different growth environments and respond differently to environmental changes such as climate change or nitrogen deposition. For example, many fungi OTUs belonging to *Cortinarius*, *Tylospora*, and *Piloderma* tended to occupy cold habitats; conversely, *Russula* and *Lactarius* were found across wider temperature ranges and in warmer habitats (*Miyamoto, Terashima & Nara, 2018*). These results indicate that ecological environmental factors significantly impact on the community structure of EM fungi. Most EM fungi have a significant correlation with ecological environment factors, which indicates the importance of EM fungi as underground indicator species of forest status and environmental conditions (*Suz et al., 2014*), and also shows the ecological specificity of EM fungi.

Overall, stochastic processes dominate the community assembly of ectomycorrhizal fungi associated with *Betula platyphylla* in Inner Mongolia, stochastic processes (37.73–98.50%) played a more important role than deterministic process (1.50–3.88%, 62.27%) in controlling community assembly of EM fungi associated with *Betula platyphylla* in five sites, in which homogenizing dispersal (2.96–27.63%) and drift (16.29–84.86%) were most important ecological processes, at the same times, there is an exception in deterministic processes, homogeneous selection accounts for 62.27% in the GH (Fig. 8). Analyzing the reasons, it is believed that the spatial scale of the investigations is one of the most important factors affecting both stochastic and deterministic processes in the formation of microbial communities (*Sun et al., 2021*), but there are also studies indicating that the relative importance of stochastic processes is time-dependent and plays a greater role in the mid stage of the community (*Zhou et al., 2014*). In our study, the deterministic processes including homogeneous selection and heterogeneous selection is not clearly manifested, which may be due to our study was conducted at a narrow scale, and the heterogeneous selection was low levels, some studies have shown that drift in stochastic processes is also an important driving factor in the assembly process of soil microbial communities (*Segre et al., 2014*; *Catano, Dickson & Myers, 2017*; *Wang et al., 2019a*; *Wang et al., 2019b*). The drift is derived from the limits of spatial distance on the movement of microbes, thus a phenomenon where the migration rate of microorganisms is higher in the same area due

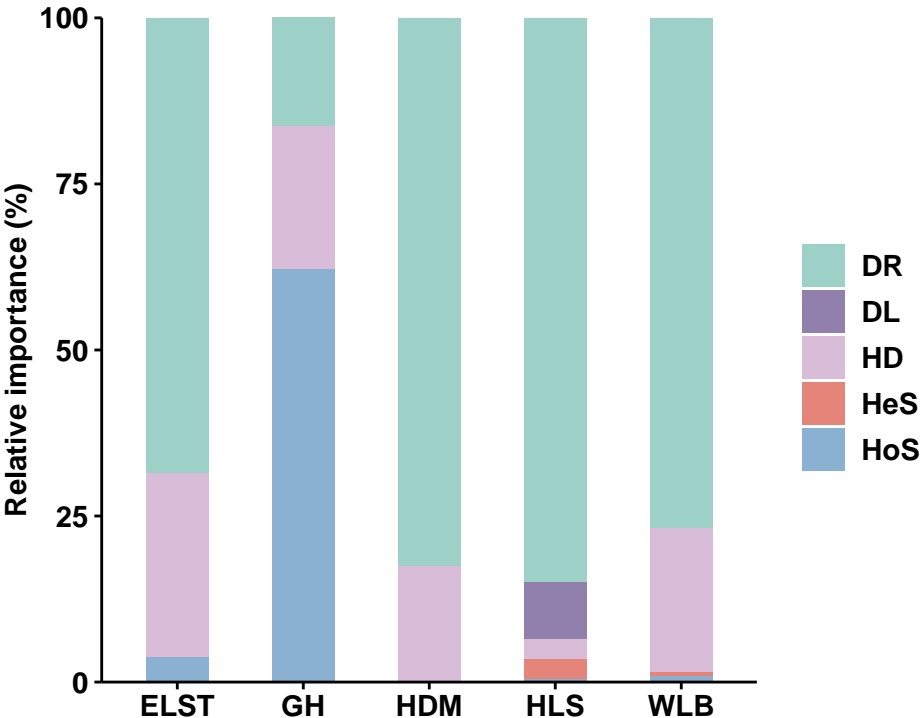

**Figure 8** Ecological processes underlying community assembly of ectomycorrhizal (EM) fungi. relative importance of ecological processes in determining EM fungal communities in each site. HoS, homogeneous selection; HeS, heterogeneous selection; HD, homogenizing dispersal; DL, dispersal limitation; and DR: drift and others.

to good environmental conditions (*Evans, Martiny & Allison, 2017*; *Ning et al., 2020*). This is also consistent with the high homogeneity distribution rate in our study.

In conclusion, the results of this study indicate that different fungi groups often have different ecological niches.

## CONCLUSIONS

A total of 2,039 fungal OTUs were obtained in this study, 295 of which were EM fungi, belonging to two phyla, three classes, nine orders, 20 families and 31 genera. It can be divided into three classes, Agaricomycetes is the absolute advantage class. *Russula*, *Cortinarius* and *Sebacina* are the dominant genera in the rhizosphere soil of *B. platyphylla*. The EM fungal community is affected differently by the soil physical and chemical properties, in which TN, TOC, AK, pH, longitude, and latitude have a very important influence on the EM fungal composition, and other factors also have significant influence. Neutral model analysis (NCM) and βNTI index found that the community structure was mainly affected by the stochastic process, which could provide a better basis for the community assembly of EM fungi. The results of this study can provide a reference for the research on the adaptability of EM fungi to the environment in different regions, and also provide a direction for the research on plant stress resistance in Inner Mongolia, China.

## ACKNOWLEDGEMENTS

We thank Yun Su from the Helan Mountain National Nature Reserve (HLS), and Xiaoliang Zhang from Genhe (GH) for their help during the sampling.

### Funding

This study was supported financially by the National Natural Science Foundation of China (32260006, 32260027, 31760169), the Science and Technology Project of Inner Mongolia Autonomous Region (no. 2019GG002), the Natural Science Foundation of Inner Mongolia Autonomous Region (2017MS0310), the Fundamental Research Funds for the Inner Mongolia Normal University (2022JBTD010), and the High-level Talents Introduced Scientifific Research Startup Fund Project of Baotou Teacher's College (No. BTTCRCQD2020-001), and the Science and Technology Project of Ordos (no. 2022YY008), and Basic Scientific Research Business Fee Project for Directly Affiliated Universities in Inner Mongolia Autonomous Region (no. 2023RCTD021). The funders had no role in study design, data collection and analysis, decision to publish, or preparation of the manuscript.

### Grant Disclosures

The following grant information was disclosed by the authors:
National Natural Science Foundation of China: 32260006, 32260027, 31760169.
Science and Technology Project of Inner Mongolia Autonomous Region: 2019GG002.
The Natural Science Foundation of Inner Mongolia Autonomous Region: 2017MS0310.
The Fundamental Research Funds for the Inner Mongolia Normal University: 2022JBTD010.
High-level Talents Introduced Scientifific Research Startup Fund Project of Baotou Teacher's College: BTTCRCQD2020-001.
Science and Technology Project of Ordos: 2022YY008.
Basic Scientific Research Business Fee Project for Directly Affiliated Universities in Inner Mongolia Autonomous Region: 2023RCTD021.

### Competing Interests

The authors declare there are no competing interests.

### Author Contributions

- Min Li conceived and designed the experiments, analyzed the data, authored or reviewed drafts of the article, and approved the final draft.
- Zhaoyun Meng conceived and designed the experiments, performed the experiments, prepared figures and/or tables, and approved the final draft.
- Jinyan Li performed the experiments, prepared figures and/or tables, and approved the final draft.

- Xuan Zhang analyzed the data, prepared figures and/or tables, authored or reviewed drafts of the article, and approved the final draft.
- Yonglong Wang conceived and designed the experiments, analyzed the data, authored or reviewed drafts of the article, and approved the final draft.
- Xinyu Li performed the experiments, prepared figures and/or tables, and approved the final draft.
- Yuze Yang performed the experiments, prepared figures and/or tables, and approved the final draft.
- Yue Li performed the experiments, prepared figures and/or tables, and approved the final draft.
- Xunjue Yang performed the experiments, prepared figures and/or tables, and approved the final draft.
- Xiuli Chen conceived and designed the experiments, analyzed the data, authored or reviewed drafts of the article, and approved the final draft.
- Yongjun Fan conceived and designed the experiments, analyzed the data, prepared figures and/or tables, authored or reviewed drafts of the article, and approved the final draft.

## Field Study Permissions

The following information was supplied relating to field study approvals (i.e., approving body and any reference numbers):

Field experiments were approved by the Helan Mountain National Nature Reserve in Inner Mongolia.

## DNA Deposition

The following information was supplied regarding the deposition of DNA sequences:

The raw sequence data reported in this paper have been deposited in the Genome Sequence Archive (Genomics, Proteomics & Bioinformatics 2021) in National Genomics Data Center (Nucleic Acids Res 2022), China National Center for Bioinformation/Beijing Institute of Genomics, Chinese Academy of Sciences (GSA: CRA013683) that are publicly accessible at https://ngdc.cncb.ac.cn/gsa/browse/CRA013683.

## Data Availability

The raw sequence data is available in the Genome Sequence Archive: CRA013683.

## Supplemental Information

Supplemental information for this article can be found online at http://dx.doi.org/10.7717/peerj.19364#supplemental-information.

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
