# Peer review of "Stochastic processes dominate the community assembly of ectomycorrhizal fungi associated with Betula platyphylla in Inner Mongolia, China"

_PeerJ, doi:10.7717/peerj.19364_

## Round 0.1 · original submission · Major Revisions

Thank you for your submission to PeerJ. Each of the reviewers commented favorably on your manuscript; however, we feel that additional information and clarifications are needed. Numerous specific suggestions for improvement are provided in the reviews. Please address each of the comments/suggestions in your revised manuscript. In addition, please provide information on how this work differs from your previously published work; how do your new results fit into what has been done previously and what new knowledge is gained through this work.

Reviewer 1 ·

Basic reporting

The English is somewhat clear, but there are lots of small issues that should be fixed. The wording could be more concise.

More literature needs to be discussed in relation to the community assembly processes. I have tried to provide specific suggestions below.

The figures look professional, and the raw data is available. However, more information is needed for some of the excel tables so that it is clear what each column contains.

The study is mostly self contained, but refers to other published studies for lots of the methods details instead of providing them. More details should definitely be provided within the manuscript itself, I have made some suggestions.

Experimental design

The experimental design seems robust, but three plots within each site was clearly not enough to capture the fungal diversity. This needs to be acknowledged more often throughout the manuscript, particularly when interpreting some of the results (see my specific comments).

Much more detail regarding the sequence processing and OTU assignments are needed, along with much more detail of the rational and methods for the statistical analyses (see specific comments below). Also, can the site map go in the actual manuscript rather than the supplementary information? It’s nice to be able to see the site layout.

The analyses seem mostly robust, but lots more detail of how they were carried out and why are needed, particularly in relation to the stochastic versus deterministic community assembly analysis. I cannot judge how robust that analysis is with the detail provided.

Validity of the findings

Most findings seem valid, except (as mentioned above) more information about the community assembly analysis is needed to understand those findings. There is no Discussion section of those findings which is a big problem. Additionally, the first part of the results discusses all the environmental variables that are impacting community composition, but the last part says assembly processes are largely stochastic. If the interpretation of the first part of the results is true then likely it is a combination of both stochastic and deterministic processes driving patterns. More detail is needed so it is clear why the authors are certain patterns are mostly driven by stochastic processes. Is there are way to calculate the relative importance of stochastic and deterministic processes? There are some percentages from the RCbray analysis (more detail is needed about what this is), but these only seem to refer to different types of stochastic processes.

In general I find the manuscript needs to be better structured so that the findings are clearer. The Results and Discussion need to be shorter and more concise, and more carefully linked back to the main research questions. I think the current Discussion needs lots of work to make it clear what the main findings are and how and why they are similar or different to past research. More discussion of past literature is needed. Additionally, the implications of the findings are not clear in the Discussion. The manuscript would benefit from more in-depth discussion about what the results mean for improving our understanding of the ecological processes operating in these forests, or what they mean for conservation or restoration.

Additional comments

I have tried to provide specific comments throughout the manuscript to help you address the issues I outlined above.

Abstract
L23: consider using “driving mechanisms” as there are likely more than one
L24: please define EM here as it’s the first time it’s mentioned
L26: delete “the” from “with the characteristics”

Introduction
In general, the introduction mentions a few times that EM have crucial roles in ecosystems, but there is no explanation of what these roles are. You should add more details of this to better justify why your study is important, as well as more details regarding why the community composition of the fungi matters and why it’s important to understand the relative importance of environmental filtering versus stochastic factors in structuring the communities.
L51: not sure what you mean by “are certain groups in mycorrhizal fungi”. Do you mean they are one of the main types of mycorrhizal fungi?
L53: not sure what you mean by “played important roles in the ecosystems”. Which ecosystems are you referring to and what are the important roles?
L55-56: this sentence is confusing and I’m not sure what you mean, please try and re write so your point is clearer.
L58: the term “important” would fit better than “significant” here
L64-66: please include some references to back up this statement
L75: better to start the Liu et al example as a new sentence
L87: delete “deserves”
L86-88: please write out what you mean by “it” and “their”
L89-92: these sentences say B platyphylla is important, but not exactly why. Please provide more explanation
Line104: high diversity compared to what? Can you list how many species were detected here, or state how high the diversity is compared to other habitat types or regions?
Line 104: do you mean research to date – or past studies? “current research” is easily confused with your research in this manuscript
Line 106: starting the sentence with “As” doesn’t fit
Line 116-119: consider rephrasing the questions to be more specific and more easily testable – perhaps something like “(1) Does EM fungal diversity and community composition change across environmental zones? (2) What is the relative importance of environmental filtering versus dispersal limitation in influencing EM fungal community assembly?”. You should also refer back to these aims in the Discussion so that it is clear what the answers to these questions are

Methods
L124: delete “along”
Line125: turn the part beginning “which selected forests…” into a new sentence because it is very long
Supplementary Fig 1: maybe I’m not looking in the right place, but I can’t find a caption for this figure anywhere. Having a site map is great, but I am interested to know what the lines dividing the landscape are. And it would be also nice to see the environmental zones described in the Introduction (maybe that’s what the lines show). Also is it necessary to have two maps showing the same thing just at different zoom levels? I think it would be better to just have one slightly bigger map, and then a small outline of the country in the corner with a square around the region where the sites are.
Line129-130: what was there 20 years ago prior to the forests? It would be good to provide this history if known because 20 years is not very long for all the legacy affects from past states to have disappeared.
L130-135: it looks like from Supplementary Table 1 that there were three plots/sampling areas selected within each of the five sites. Please explain that here, and make it clear whether the 4-7 trees were taken within each plot (therefore 12-21 trees at each site in total) or if there were 4-7 trees within the whole site. Perhaps the exact number of trees from each plot along with some more detailed site descriptions could be added to Supplementary Table 1
L145: replace “should be” with “were”
Line161: please add more details of how you processed the raw sequences here rather than referring to past papers. How did you make sure errors were removed? What software did you use? How did you group the OTUs? How did you adjust for uneven read depths between samples? How did you assign the OTUs names so you could decide which ones were EM species?
L172: delete “In our study”. Also is there a reference for the Krona chart software? And I don’t see the Krona chart as a figure – so do you really need to mention it here?
L176: “which was based on”
L173-180: this sentence is too long and hard to follow. Please split it up to talk about each type of analysis separately with a bit more detail, and why it was that you performed each one. Particularly what the NCM and MNTD analysis are trying to test. This information is essential for understanding your results so I think it needs to be explained here rather than just referring to past papers. If there is not space in the word count perhaps you can add a supplementary methods document.

Results
L187: I’m not sure what “de-redundancy and chimera” means
L190: what does “and flattening of EM fungi for analysis” mean?
Supplementary Table 2: great table, but you need a “read me” tab or detailed table caption explaining what all the columns are and what the values area (are they raw read counts or something else?)
Figure 1: Is the sample size of each block 3 (the 3 plots in each site)? Please make this clear in the caption. Also I don’t think you mean the adonis test here – please check
L198-200: I don’t think you can say this if the accumulation curves were not flat – because lots of the “rare” species may actually be quite common if you were to sample the site in more depth. I suggest deleting this sentence
L201-204: you say you used a t-test, but then the next sentence says it was a Wicoxon test. Did you do both or just one? Also why didn’t you do an ANOVA instead of t-tests considering there are five sites?
L204: are these the means across the full data set? The previous sentence said the values differed between sites, so I’m not sure it makes sense to present the whole population mean and not look at the site means individually.
L205: Can you refer to “site ELST” so it is obvious what this is? When I first read it I thought it was another statistic of some sort
L206-209: This is a good summary, but a repeat of the above sentences. I like this sentence better – can you just have this with the statistics in brackets, and remove the above sentences?
L219: “detected at GH” – and change “of” to “at” in other places as well
L221-224: this sentence doesn’t add much – I’d suggest rewriting so the result is included. Is the pattern the same for the genera as what was just described for the lineages, or is it different? Although figure 2 looks like it’s about the lineages, not genera. So perhaps delete this sentence and reference Fig 2 above.
L225-230: these results are essentially about the fungal richness at each site, which was already discussed in the section above. Can these numbers be moved up there, and this section be kept to just discussing the shared/different OTUS in the venn disgrams?
L232-234: I don’t think the venn diagram shows that. It doesn’t give any information about the reasons for the different OTUs in each site, you need other more detailed analysis for that.
L234-235: I don’t think you can say this. Your species accumulation curves were not flat – so how do you know that all the fungal species wouldn’t be found at all the sites if you were to sample them in more depth? The sampling depth problem needs to be mentioned here. Because the sample size was the same at each site (and therefore each site is biased equally) I think it’s ok to compare the number of unique OTUs between sites (e.g. HDM has a lot less than the other sites). But it needs to be acknowledged that these OTUs might not be truly unique. And other environmental analyses are needed before you can conclude that it is due to environmental conditions (what if it is due to difference in the site histories?).
Figure 4: it looks like there are both big points and small points on this figure. What do they mean? If the big points are just group centroids I think they should be removed because they cover up the small points.
Figure 5: I like having the environmental vectors on the figure. But why did you do both a PCoA and an MDS? I think the MDS is enough. If not, please explain in the Methods why you needed both.
L244-255: I think this is better suited to the discussion, unless you are talking about specific results presented in the figure.
L254-255: can you really say this without doing an analysis including other factors relating to topography, site history etc?
Line256: please explain what these statistics are. Why does that result mean that phylogenetic relatedness in the communities was not stochastic? Also what was the actual p value (was it close to 0.05, or way bigger)?
L260-261: I’m not familiar with methods to enough to know why that means communities are mainly affected by stochastic processes. What would the result look like if deterministic factors were driving patterns? More details are needed here to follow what the results are showing.

Discussion
L271-282: this paragraph could be rewritten to more clearly recap the aims and key results of the study, and why the results are useful and exciting. The first sentence is already in the results so the exact numbers are not needed again here.
L287: change “fungal” to “fungi”
L283-297: I don’t think that the exact differences between the roles of Basidiomycota and Ascomycota is clear here. Both come across as decomposers that can compose lignin, I think the differences could be made clearer. The information is also quite general and not linked well with the results of the study. I think this paragraph could be combined with the following paragraph, and talk more about the functions performed by the different genera (rather than the phyla).
L309: “fungi” instead of “fungal” – in other locations as well
L269: I think this section is lacking depth in general, and could be shortened and made more specifically about the effects of the differences in community composition at each of the sites examined (e.g. how nutrient cycling and decomposition might be affected at different sites based on what we know about what different fungal genera do). There is also no discussion of how the communities compare to other studies in these forests or nearby regions (those local scale ones mentioned in the Introduction).
L362-363: Is this presented in the results somewhere? I don’t remember an analysis of how abundance changes along different environmental gradients. Please make sure that all patterns discussed in this section are based on analyses presented in the results
L381-382: this is said multiple times throughout this section. I think this section could be made shorter and more concise so you can conclude this just once at the end.
L392: Why is there no Discussion section discussing the community assembly analyses? This is what the title of the manuscript is based on, and one of the key questions in the Introduction. A large part of the Discussion needs to be dedicated to this. There are lots of past studies showing different patterns, including that deterministic processes drive EM community assembly in many places. These need to also be discussed along with why the sites in this study might be showing stochastic patterns. The following papers might be relevant, but there are many more:
https://besjournals.onlinelibrary.wiley.com/doi/full/10.1111/1365-2745.13910
https://onlinelibrary.wiley.com/doi/full/10.1111/mec.16860
https://onlinelibrary.wiley.com/doi/full/10.1111/mec.14414
https://www.sciencedirect.com/science/article/pii/S0038071714003095
L394-395: this is the fourth time these numbers are provided, please focus on the overall conclusions of the paper rather than repeating the results.

·

Basic reporting

No comment

Experimental design

No comment

Validity of the findings

No comment

·

Basic reporting

The authors studied Stochastic processes dominate the community
assembly of ectomycorrhizal fungi associated with Betula platyphylla in Inner Mongolia, China. They found that there were significant differences in the composition of dominant genera of EM fungi across the five sites, and the relative abundances of dominant genera also showed significant differences among the sites.
Overall, I enjoyed reading the manuscript and support the publication after revision although I had some comment which may help improve this manuscript.

Experimental design

It would be better to add more detail on how the authors performed analysis of soil properties, and bioinformatics analysis.
How many replicates the authors had for each site? βMNTD was more widely used to differentiated four different ecological processes such homogenization selection, heterogeneity selection and so on, which could provide more detail information for soil community assembly. However, the estimation requires sufficient biological replicates (e.g., ≥ 6) to determine βMNTD.

Can you explain why soil but not plant roots was used to extract DNA and sequence?

Validity of the findings

I would suggest the authors add a short section to the discussion to confirm that their results will not depend on seasonal variation since many studies showed that temporal variation was more important in determining soil community assembly.

Additional comments

Line 187: the sentence was not clear. These 377,969 sequences were for all soil samples or each sample from each site?

---

## Round 0.2 · Major Revisions

Thank you for your re-submission of the manuscript. I appreciate the author's efforts to address the reviewer's previous comments, and as a result of those efforts, I do feel the context of the study is clearer to the reader. However, I agree with the reviewer's assessment that the manuscript still requires substantial revisions. In particular, in your revision, please fully address *each* of the concerns raised by both reviewers with bullet point-by-point responses and also make the necessary changes to the manuscript. For example, more information is needed for supplementary tables, methods, and missing references. Since the stated main goal of the study is to determine the ecological processes that structure communities at larger spatial scales, I agree with Reviewer 1 that substantially more information is needed for the methods (including how you determined fungal taxonomy and which OTU were EcM; rarefaction of samples, etc.), the methods and rationale for the statistical analyses (e.g., were data analyzed per tree or per site?), and how to interpret the incongruence between different statistical analyses on deterministic vs. stochastic mechanisms for community assembly. Lastly, as shown now in tables and figures, the fungal taxonomic names are incorrect (e.g., lowercase). I look forward to your revised manuscript. Thank you for your efforts.

Reviewer 1 ·

Basic reporting

I do not feel that the authors have responded adequately to the comments. They only select some of the comments to include in their response document. Many comments are missing, and looking at the reviewed manuscript, they have not considered many of them when doing the revisions.

Additionally – the authors say they have provided more detail in the supplementary files, but many columns still have no explanation. E.g., what do the acronyms in the column names of Supplementary Tables 1, 2 and 3 stand for? All files should have a readme tab that explains each column.

I’m still concerned that details of the methods are not provided in the manuscript itself. I have tried to find the reference cited for the bioinformatics methods (Fan et al. (2023)), but only a Fan et al. (2013) is listed in the reference list. And when I look in the issue and volume of the journal cited for Fan et al. (2013) I cannot find an article from those authors or with that title.

Experimental design

I’m still missing details and rational for the statistical analyses. It’s not enough to cite that methods are described in past papers. And limitations of the experimental design are still not well acknowledged.

Validity of the findings

I cannot judge the validity of the findings with the information provided. As commented during the last round of review, much more information is needed.

·

Basic reporting

no comment

Experimental design

no comment

Validity of the findings

no comment

Additional comments

no comment

---

## Round 0.3 · accepted · Accept

While the content and writing are now OK, the language still needs to be improved.

Reviewer 1 ·

Basic reporting

The authors have now responded to all my comments from the first round of review, so thank you. But I cannot find their response to my comments for the second round of review.

However, the addition of bioinformatics and more statistical analysis methods is helpful and satisfies some of my concerns.

The authors still have not provided more details about what the columns are in the supplementary Tables. The authors either need to provide explanations of all acronyms or explain why they disagree with this request. Also, Supplementary Table 3 looks like it has an accidental extra tab.

Some of the new discussion material feels like it should go in the results (e.g., lines 530-537).

Experimental design

-

Validity of the findings

-

Additional comments

-